# Identification of a carbohydrate recognition motif of purinergic receptors

Lifen Zhao[1], Fangyu Wei[1,2], Xinheng He[1,2], Antao Dai[1], Dehua Yang[1,2], Hualiang Jiang[1,2,3], Liuqing Wen[1,2]*, Xi Cheng[1,2,3]*

[1]State Key Laboratory of Drug Research, Carbohydrate-Based Drug Research Center and National Center for Drug Screening, Shanghai Institute of Materia Medica, Chinese Academy of Sciences, Shanghai, China; [2]University of Chinese Academy of Sciences, Beijing, China; [3]School of Pharmaceutical Science and Technology, Hangzhou Institute of Advanced Study, Hangzhou, China

**Abstract** As a major class of biomolecules, carbohydrates play indispensable roles in various biological processes. However, it remains largely unknown how carbohydrates directly modulate important drug targets, such as G-protein coupled receptors (GPCRs). Here, we employed P2Y purinoceptor 14 (P2Y14), a drug target for inflammation and immune responses, to uncover the sugar nucleotide activation of GPCRs. Integrating molecular dynamics simulation with functional study, we identified the uridine diphosphate (UDP)-sugar-binding site on P2Y14, and revealed that a UDP-glucose might activate the receptor by bridging the transmembrane (TM) helices 2 and 7. Between TM2 and TM7 of P2Y14, a conserved salt bridging chain ($K^{2.60}$-$D^{2.64}$-$K^{7.35}$-$E^{7.36}$ [KDKE]) was identified to distinguish different UDP-sugars, including UDP-glucose, UDP-galactose, UDP-glucuronic acid, and UDP-$N$-acetylglucosamine. We identified the KDKE chain as a conserved functional motif of sugar binding for both P2Y14 and P2Y purinoceptor 12 (P2Y12), and then designed three sugar nucleotides as agonists of P2Y12. These results not only expand our understanding for activation of purinergic receptors but also provide insights for the carbohydrate drug development for GPCRs.

*For correspondence:
lwen@simm.ac.cn (LW);
xicheng@simm.ac.cn (XC)

Competing interest: The authors declare that no competing interests exist.

## Editor's evaluation

This study describes a valuable model for the interaction of nucleotides, an important group of signalling molecules in health and disease, with their receptors. By combining experimental and theoretical methods, the authors provide compelling evidence for the mechanism of receptor activation. The results will be useful for drug design to target these receptors.

## Introduction

As significant components of the organism, carbohydrates play indispensable roles in energy supply, cell signaling, and immune responses (*Gagneux and Varki, 1999*). Dysregulation of carbohydrates has been proved to be associated with the development of various diseases (*Reily et al., 2019*). However, it is still elusive how carbohydrates directly act on major therapeutic targets, including G-protein coupled receptors (GPCRs) (*Cheng and Jiang, 2019*; *Hauser et al., 2017*). P2Y purinoceptor 14 (P2Y14) represents an outstanding model system for understanding carbohydrate modulation of GPCRs. It belongs to P2Y purinoceptor subfamily, consisting of receptors responding to nucleotides, including adenosine diphosphate (ADP) and UDP (*Ralevic and Burnstock, 1998*). Distinct from the other purinoceptors, P2Y14 is potently activated by UDP and a class of carbohydrates, that is, UDP-sugars (*Abbracchio et al., 2006*; *Jacobson et al., 2020*). UDP-sugars activate P2Y14 with a relative potency order of UDP-glucose (UDP-Glc), UDP-galactose (UDP-Gal), UDP-glucuronic acid (UDP-GlcA),

**eLife digest** Sugars and other types of carbohydrates are biomolecules which play a range of key roles in the body. In particular, they are important messengers that help to coordinate immune responses. For example, a carbohydrate known as UDP-Glucose (a kind of UDP-sugar) can activate P2Y14, a receptor studded through the surface of many cells; this event then triggers a cascade of molecular events associated with asthma, kidney injury and lung inflammation. Yet it remains unclear how exactly UDP-Glucose recognizes P2Y14 – and, more broadly, how carbohydrates interact with purinergic receptors, the class of proteins that P2Y14 belongs to.

To examine this question, Zhao et al. combined functional experiments in the laboratory with molecular dynamics simulations, a computational approach. This work revealed that UDP-Glucose may activate P2Y14 by bridging its segments anchored within the cell membrane. A component of P2Y14, known as the KDKE chain, was found to have an important role in distinguishing between highly similar types of UDP-sugars. This allowed Zhao et al. to design three sugar molecules which could activate another purinergic receptor that also contained a KDKE chain.

Purinergic receptors are promising therapeutic targets. A finer understanding of how they recognise the molecules that activate them is therefore important to be able to identify and design new drug compounds.

and UDP-*N*-acetylglucosamine (UDP-GlcNAc) (*Chambers et al., 2000*; *Hamel et al., 2011*; *Ko et al., 2009*; *Ko et al., 2007*). These sugar nucleotides act as important signaling molecules via P2Y14 to mediate many physiological processes (*Amison et al., 2017*; *Breton and Brown, 2018*; *Ferreira et al., 2017*; *Lazarowski, 2010*; *Müller et al., 2005*; *Sesma et al., 2016*). Particularly, UDP-Glc regulates immune responses and associate with asthma, kidney injury, and lung inflammation (*Amison et al., 2017*; *Breton and Brown, 2018*; *Ferreira et al., 2017*; *Müller et al., 2005*; *Sesma et al., 2016*). As an isomer of UDP-Glc, UDP-Gal is present in various cell models, including physiologically relevant primary cultures of human bronchial epithelial cells (*Lazarowski, 2010*). It remains unknown how these sugar nucleotides are recognized by P2Y14.

As the closest homolog to P2Y14, P2Y purinoceptor 12 (P2Y12) has not been reported to be activated by any sugar nucleotide (*Jacobson et al., 2020*; *Ralevic and Burnstock, 1998*). P2Y12 is potently activated ADP. The reported agonist-bound structures of P2Y12 provide insights to understand the nucleotide activation of P2Y purinoceptors. The crystal structures of P2Y12 show that a full agonist 2-methylthio-adenosine-5′-diphosphate (2MeSADP, a close analogue of ADP) binds to an extracellular pocket consisting of transmembrane (TM) helices and extracellular loops (*Zhang et al., 2014a*). Since P2Y12 is highly similar to P2Y14 with 45.67% amino acid sequence identity, it would be interesting to investigate whether this receptor is also sensible to sugar nucleotides.

Here, we combined molecular docking, molecular dynamics (MD) simulations, and functional study to reveal the molecular mechanism how P2Y14 is activated by a sugar nucleotide. The ligand-binding models of different UDP-sugars (UDP-Glc, UDP-Gal, UDP-GlcA, and UDP-GlcNAc) were quantitatively characterized to identify the sugar recognition site of P2Y14. Both P2Y14 and P2Y12 were employed to unveil a conserved sugar-binding motif. Multiple carbohydrates were designed and validated as their agonists targeting the conserved functional motif.

## Results

### Identification of sugar-binding site in P2Y14

Both UDP and UDP-Glc potently activate P2Y14 with EC50 values of 50.9±6.1 nM and 40.3±1.5 nM, respectively (*Figure 1A–C*). Compared with UDP, UDP-Glc showed an increased potency on P2Y14 at high concentration (*Figure 1B–C*), suggesting that the sugar moiety of UDP-Glc contributes to activating P2Y14. To investigate how UDP-Glc regulates the P2Y14 via its sugar moiety, we used molecular docking to construct UDP-Glc-bound models of P2Y14 and compared them with UDP-bound P2Y14 models (*Figure 1D–G*). Because the protein structure of P2Y14 is unrevealed, we employed the X-ray structures of P2Y12 (*Zhang et al., 2014a*) as templates to constructed homology models of human P2Y14. The molecular docking showed that both UDP and UDP-Glc bind to an extracellular

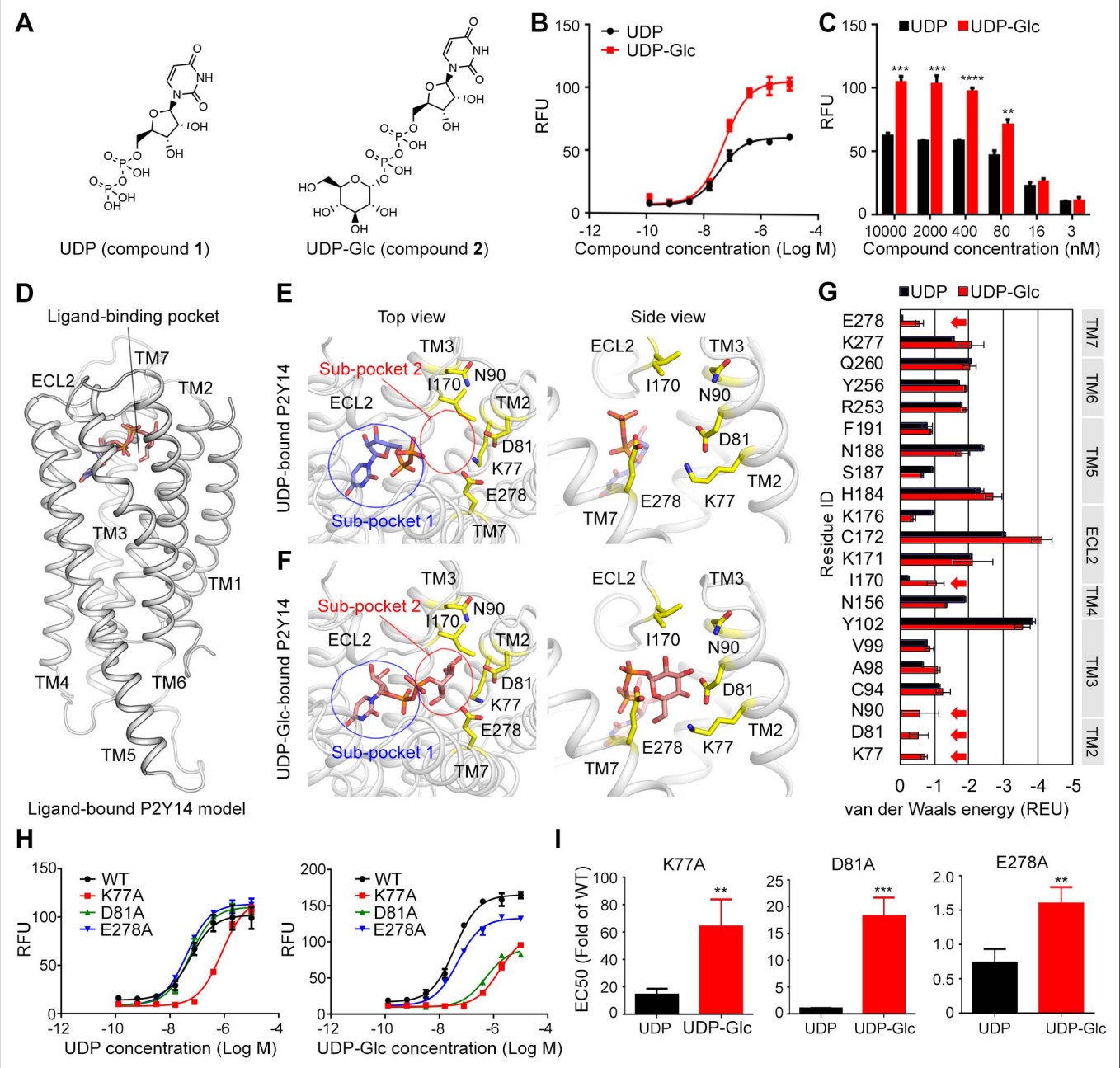

**Figure 1.** Identification of sugar-binding sites in P2Y purinoceptor 14 (P2Y14). (**A**) Chemical structures of uridine diphosphate (UDP) and UDP-glucose (UDP-Glc). (**B**) Concentration-response curves of calcium mobilization for UDP or UDP-Glc in HEK293 cells transiently co-transfected with human P2Y14 and Gα$_{qi5}$. Data are shown as mean ± SEM (n=3). See also *Figure 1—source data 1* and *Supplementary file 1*. (**C**) Concentration-dependent comparison of calcium mobilization for UDP and UDP-Glc in HEK293 cells transiently co-transfected with human P2Y14 and Gα$_{qi5}$ (n=3); **p<0.01, ***p<0.001, ****p<0.0001. (**D**) Ligand-bound model of P2Y14. Protein and compound are shown in cartoon and stick representations. (**E, F**) Docking models of UDP (**E**) and UDP-Glc (**F**) to P2Y14. Key residues are highlighted in yellow. Two sub-pockets for ligand binding are marked with circles. (**G**) Decomposition of ligand-binding energy for each receptor residue (n=10). (**H**) Calcium mobilization concentration-response curves for UDP or UDP-Glc in HEK293 expressing P2Y14 wild-type (WT) and mutants (n=3). See also *Figure 1—source data 2* and *Supplementary file 1*. (**I**) Comparison of EC50s for UDP-Glc and UDP in HEK293 cells expressing P2Y14 mutants in calcium mobilization assay (n=3); **p<0.01, ***p<0.001. See also *Figure 1—source data 3*.

The online version of this article includes the following source data and figure supplement(s) for figure 1:

**Source data 1.** Potency of uridine diphosphate (UDP) or UDP-glucose (UDP-Glc) in HEK293 cells expressing P2Y purinoceptor 14 (P2Y14).

**Source data 2.** Potency of uridine diphosphate (UDP) or UDP-glucose (UDP-Glc) in HEK293 cells expressing P2Y purinoceptor 14 (P2Y14) wild-type (WT)

*Figure 1 continued on next page*

*Figure 1 continued*

and mutants.

**Source data 3.** Comparison of EC50s for uridine diphosphate-glucose (UDP-Glc) and UDP in HEK293 cells expressing P2Y purinoceptor 14 (P2Y14) mutants.

**Figure supplement 1.** The sub-pocket 1 of P2Y purinoceptor 14 (P2Y14) for uridine diphosphate (UDP) and UDP-glucose (Glc).

pocket consisting of TM helices 2–7 and extracellular loop (ECL) 2 (*Figure 1D*), which is corresponding to a known agonist binding pocket of P2Y12 (*Zhang et al., 2014a*). The docking score of UDP-Glc is –9.3 kcal/mol, and that of UDP is –8.5 kcal/mol, indicating that both UDP and UDP-Glc bind to P2Y14.

Compared with the UDP-bound receptor model (*Figure 1E*), the UDP-Glc-bound model showed extra interactions between the glucose moiety and the TM2, TM3, TM7, and ECL2 of P2Y14 (*Figure 1F*), enhancing the binding of UDP-Glc. Based on these molecular docking models, we further decomposed the ligand-binding energy to each receptor residue (*Figure 1G*). Five residues (K77$^{2.60}$, D81$^{2.64}$, N90$^{3.21}$, I170$^{ECL2}$, and E278$^{7.36}$; superscript indicates Ballesteros-Weinstein residue numbering; *Ballesteros and Weinstein, 1995*) were predicted to stabilize UDP-Glc binding (*Figure 1F and G*), while they made few energetic contributions (van der Waals energy >–0.25 Rosetta energy unit) to UDP binding (*Figure 1E and G*). As shown in *Figure 1E and F*, two sub-pockets of P2Y14 were unveiled for ligand binding. The sub-pocket 1 is formed by 16 residues of TMs 3–7 and ECL2 (*Figure 1G*, *Figure 1—figure supplement 1A,B*) and binds to the nucleotide moiety of the agonist, that is, UDP. The sub-pocket 2 is the specific sugar-binding site involving K77$^{2.60}$, D81$^{2.64}$, N90$^{3.21}$, I170$^{ECL2}$, and E278$^{7.36}$ (*Figure 1F and G*). These residues are primarily charged or polar amino acids, which could made hydrogen bonds with the glucose hydroxyl groups of UDP-Glc (*Figure 1F*). To validate the proposed sugar-binding sites, we designed single-point mutations of these five residues (K77A, D81A, N90A, I170A, and E278A). Among these mutants, D81A showed significantly reduced activities by UDP-Glc compared with the wild-type (WT) group (*Figure 1H*). However, substitution of D81$^{2.64}$ by alanine did not significantly affect the receptor activities by UDP (*Figure 1H*). Interestingly, K77A mutation diminished both UDP-Glc- and UDP-induced calcium mobilization (*Figure 1H*), but it showed greater impact on UDP-Glc-induced receptor responses than UDP-induced ones (*Figure 1H, I*), suggesting extra interactions between K77$^{2.60}$ and sugar moiety of UDP-Glc. Two mutations on TM3 and ECL2 (N90A and I170A) did not significantly affect the receptor responses by UDP or UDP-Glc (*Figure 1—figure supplement 1C, D*). These findings indicate that sub-pocket 2 residues of TM2 provide major contributions to stabilizing the sugar moiety of UDP-Glc.

## UDP-Glc as a 'glue' for P2Y14 activation

The molecular docking employs rigid side chains of the receptor and does not include the influence of explicit environment on molecular interactions. To investigate how UDP-Glc interacts with P2Y14, we performed all-atom MD simulations of the P2Y14 receptor with and without UDP-Glc (*Figure 2A*). We used the molecular docking model of P2Y14 to construct the simulation systems. Each system was replicated to performed three independent simulations (see Materials and methods for detailed information). Apo P2Y14 and UDP-Glc-bound P2Y14 simulation models showed different conformations in TM6 and TM7 (*Figure 2A and B*). In UDP-Glc-bound P2Y14 simulations, the extracellular tip of TM6 shifted over 3 Å and TM7 over 4 Å toward the receptor core, compared with the apo P2Y14 simulations (*Figure 2A and B*). This inward shift of TM6 and TM7 allowed formation of polar and ionic interactions with the UDP-Glc (*Figure 2B and C*). During UDP-Glc-bound P2Y14 simulations, two charged residues K277$^{7.35}$ and E278$^{7.36}$ formed hydrogen bonds with the glucose 6' hydroxyl group of UDP-Glc to keep TM7 close to the receptor core (*Figure 2B and C*), and an arginine residue (R253$^{6.55}$) formed a salt bridge with the phosphate group of UDP-Glc to stabilize the inward shift of TM6 (*Figure 2B and C*). Consistently, compared with WT group (EC50 of 40.3±1.5 nM), single-point mutations (R253A and E278A) of TM6 and TM7 helices resulted to diminished UDP-Glc-induced calcium mobilization (EC50 of 808.6±43.6 nM for R253A and 60.2±3.6 nM for E278A) (*Figure 1H*, *Figure 2D*). In addition, at the extracellular side, the distance between TM5 and TM6 of UDP-Glc-bound P2Y14 was 5.9 Å shorter than that in the apo system (*Figure 2—figure supplement 1A, B*). Y189$^{5.41}$ and T257$^{6.59}$ made stable hydrophobic interactions to maintain the tight compact between TM5 and TM6 in UDP-Glc-bound-P2Y14 simulations, while TM6 did not interact with TM5 at the

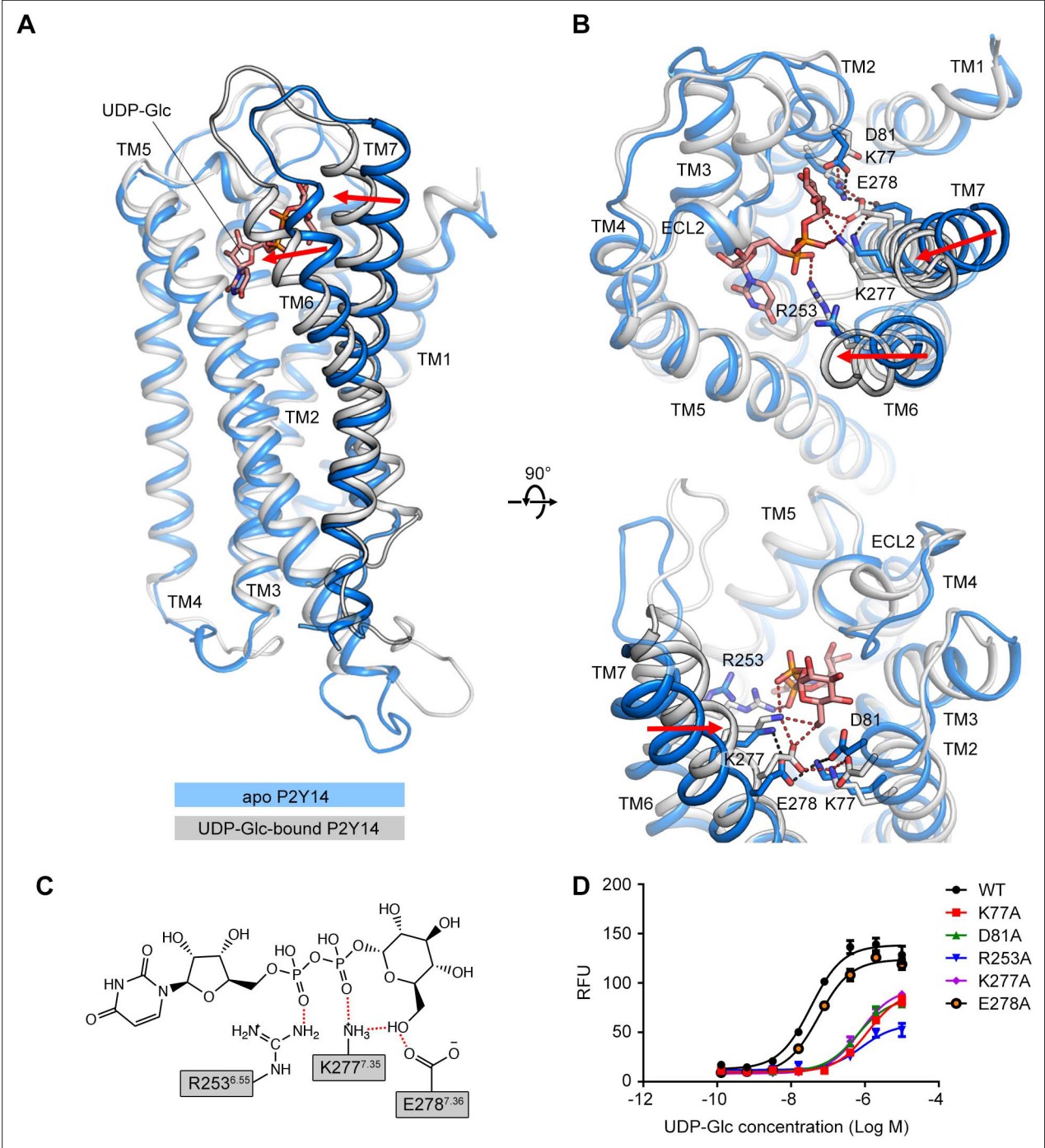

**Figure 2.** Comparison of the apo P2Y purinoceptor 14 (P2Y14) and uridine diphosphate-glucose (UDP-Glc)-bound P2Y14 simulation systems. (**A–B**) Side (**A**) and top (**B**) views of representative models of apo P2Y14 and UDP-Glc-bound P2Y14. Receptor is shown as cartoon. Ligand and key residues are shown as sticks. Movement of the extracellular tips of transmembrane (TM)6 and TM7 toward the receptor core is shown by arrows. See ***Supplementary file 2*** for computational characterization of conformational changes. (**C**) Key interactions between P2Y14 and UDP-Glc. Hydrogen bonds and salt bridges are displayed as red dashed lines. See ***Supplementary file 2*** for pairwise interaction details. (**D**) Concentration-response curves of calcium mobilization for UDP-Glc in HEK293 expressing P2Y14 wild-type (WT) and mutants. Data are shown as mean ± SEM (n=3). See also ***Figure 2—source data 1*** and ***Supplementary file 1***.

The online version of this article includes the following source data and figure supplement(s) for figure 2:

**Source data 1.** Potency of uridine diphosphate-glucose (UDP-Glc) in HEK293 expressing P2Y purinoceptor 14 (P2Y14) wild-type (WT) and mutants.

**Figure supplement 1.** Transmembrane (TM)6 orientation in apo P2Y purinoceptor 14 (P2Y14) and uridine diphosphate-glucose (UDP-Glc)-bound P2Y14 simulation systems.

extracellular side in the apo simulations (*Figure 2—figure supplement 1A, B*). Compared with WT group (EC50 of 40.3±1.5 nM), a mutation of TM6 (T257A) showed significantly reduced UDP-Glc-induced responses (EC50 of 504.9±15.9 nM) (*Figure 2—figure supplement 1C*), fully agreeing with our simulation models. Collectively, these data suggest that UDP-Glc might serve as intramolecular 'glue' to make a tight helical bundle of P2Y14, involving TM6 and TM7.

## Molecular recognition of P2Y14 via sugar-binding site

P2Y14 could be activated by different UDP-sugars with distinct potencies. With only one group substitution at the sugar moiety, UDP-Glc induced stronger activity on P2Y14 (EC50=40.3 ± 1.5 nM) than the other UDP-sugars, that is, UDP-Gal (EC50=78.3 ± 9.2 nM), UDP-GlcA (EC50=59.9 ± 4.8 nM), and UDP-GlcNAc (EC50=184.4 ± 11.8 nM) (*Figure 3A and B*). To investigate how P2Y14 recognizes different sugar moieties, we performed MD simulations of the human P2Y14 receptor complex with UDP-Gal, UDP-GlcA, and UDP-GlcNAc, respectively, and compared them with the UDP-Glc-bound P2Y14 simulations. We observed that UDP-Gal, UDP-GlcA, and UDP-GlcNAc bound to P2Y14 at the same pocket as UDP-Glc. Similar to UDP-Glc, their uridine groups occupied the sub-pocket 1 of P2Y14, while their sugar moieties bound to the sub-pocket 2 during simulations (*Figure 3C–F*, *Figure 3—figure supplements 1 and 2*). At the sub-pocket 2, a stable salt bridging chain formed by four charged residues (K77$^{2.60}$, D81$^{2.64}$, K277$^{7.35}$, and E278$^{7.36}$) were observed in all systems (*Figure 3C–F*). The negative charged glutamic acid residue E278$^{7.36}$ linked TM2 and TM7 helices by forming salt bridges with K77$^{2.60}$ and K277$^{7.35}$, while the other negative charged residue D81$^{2.64}$ forming a salt bridge with K77$^{2.60}$ to further stabilize these ionic interactions (*Figure 3C–F*).

In simulations, different sugar moieties bound to the K77$^{2.60}$-D81$^{2.64}$-K277$^{7.35}$-E278$^{7.36}$ salt bridging chain with distinct binding modes (*Figure 3C–F*, *Figure 3—figure supplement 3*). For UDP-Glc, both K277$^{7.35}$ and E278$^{7.36}$ could form hydrogen bonds with the glucose 6' hydroxyl group to keep TM7 close to the receptor core (*Figure 3C*, *Figure 3—figure supplement 4A*). However, in UDP-GlcA-bound P2Y14 simulations, at the corresponding position, the 5' carboxyl group of the sugar moiety failed to form hydrogen bond with the negatively charged E278$^{7.36}$ (*Figure 3D*, *Figure 3—figure supplement 4B*). Compared with that of WT group (EC50=59.9 ± 4.8 nM), the single-point mutation of E278A did not decrease the UDP-GlcA-induced calcium mobilization (EC50=38.2 ± 2.2 nM) (*Figure 3G*), supporting with the proposed sugar-binding model (*Figure 3D*). Substitution of 2' hydroxyl group by an acetamido group led to a rotation of the sugar moiety of UDP-GlcNAc in simulations (*Figure 3E*). Consequentially, the 6' hydroxyl of *N*-acetylglucosamine group flipped to form hydrogen bonds with K77$^{2.60}$ and D81$^{2.64}$ instead of K277$^{7.35}$ and E278$^{7.36}$ (*Figure 3E*, *Figure 3—figure supplement 4C*). Consistently, single-point mutation of D81A made significant effect to reduce the UDP-GlcNAc-induced receptor activities (*Figure 3H*). Compared with the other three UDP-sugars, UDP-Gal has a different orientation of 4' hydroxyl group. The 4' hydroxyl group of galactose formed a stable hydrogen bond with K77$^{2.60}$ and disrupted the interaction between 6' hydroxyl group with E278$^{7.36}$ (*Figure 3F*, *Figure 3—figure supplement 4D*). Compared with UDP-Glc, UDP-Gal had more interactions with TM2 and less interactions with TM7 (E278$^{7.36}$). Substitution of E278$^{7.36}$ by alanine did not significantly affect the UDP-Gal-induced receptor response (*Figure 3I*), agreeing with the proposed UDP-Gal-binding model (*Figure 3F*). For all UDP-sugars, at least three residues of K77$^{2.60}$, D81$^{2.64}$, K277$^{7.35}$, and E278$^{7.36}$ participated in ligand binding (*Figure 2*, *Figure 3*). Both computational models and experimental data indicate the K77$^{2.60}$-D81$^{2.64}$-K277$^{7.35}$-E278$^{7.36}$ salt bridging chain as a sugar-binding site of P2Y14, which can recognize different sugar moieties. The interactions of ligands with the TM7 might determine the ligand potency on P2Y14.

## Conserved sugar-binding motif for P2Y12 and P2Y14

In previous sections, we have identified K77$^{2.60}$-D81$^{2.64}$-K277$^{7.35}$-E278$^{7.36}$ salt bridging chain as an important functional site for sugar moiety recognition and UDP-sugar activation of P2Y14. These four residues (K$^{2.60}$, D$^{2.64}$, K$^{7.35}$, and E$^{7.36}$) are conserved between P2Y14 and its closest homolog, that is, P2Y12 (*Figure 4A*, *Figure 4—figure supplement 1A*). Therefore, we assumed that P2Y12 can also be activated by carbohydrate ligands. P2Y12 is activated by ADP (*Herbert and Savi, 2003*), but it has not been reported to be activated by any sugar nucleotide. To validate our assumption, we designed and synthesized three ADP-sugars, that is, ADP-glucose (ADP-Glc), ADP-glucuronic acid (ADP-GlcA), and ADP-mannose (ADP-Man), and then tested whether they can activate P2Y12 (*Figure 4B and*

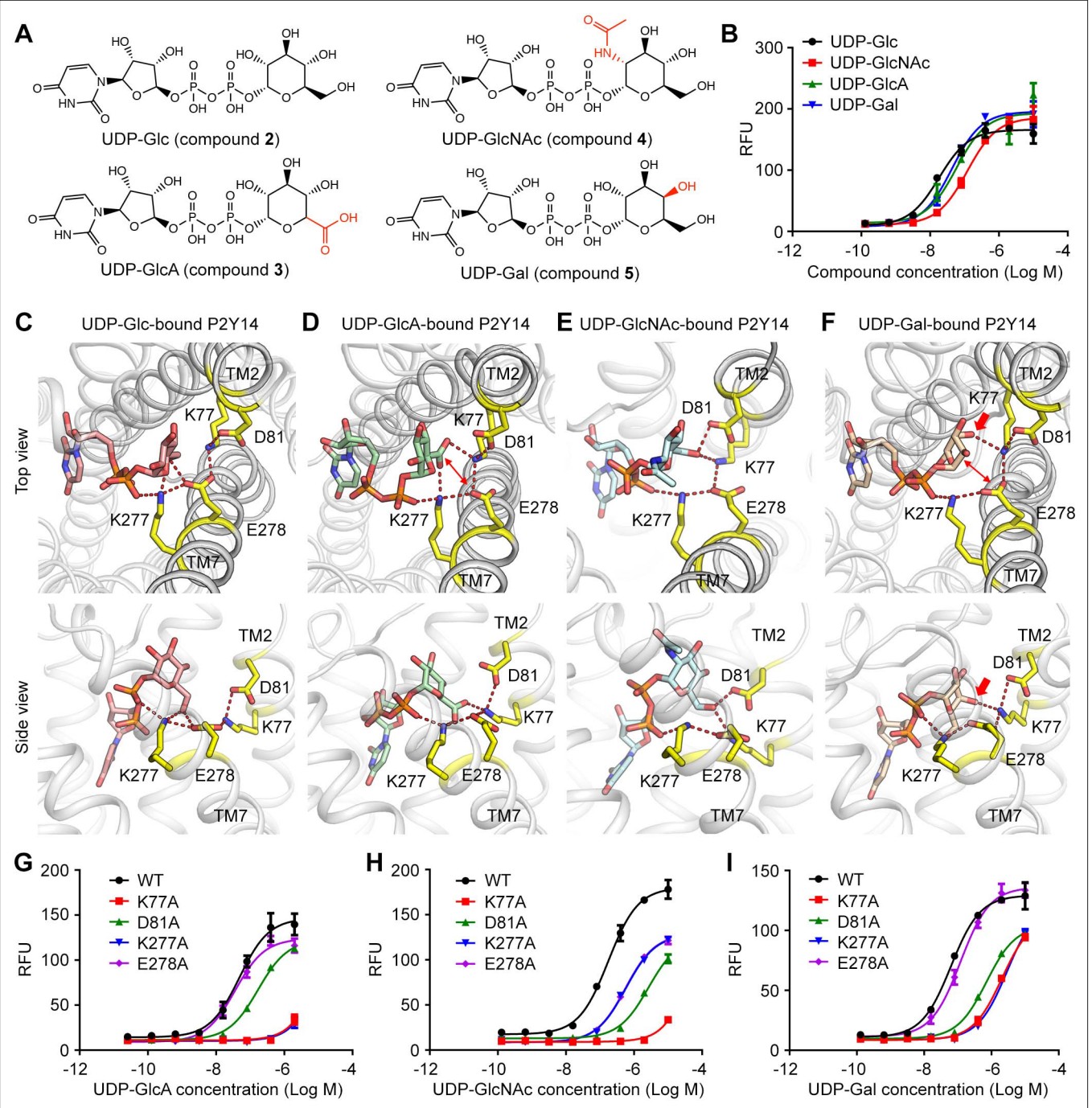

**Figure 3.** Sugar moiety recognition of P2Y purinoceptor 14 (P2Y14). (**A**) Chemical structures of uridine diphosphate-glucose (UDP-Glc), UDP-glucuronic acid (GlcA), UDP-*N*-acetylglucosamine (GlcNAc), and UDP-galactose (Gal). (**B**) Concentration-response curves of calcium mobilization for different UDP-sugars in HEK293 cells transiently co-transfected with human P2Y14 and Gα$_{qi5}$. Data are shown as mean ± SEM (n=3). See also *Figure 3—source data 1*. (**C–F**) Molecular recognition of P2Y14 for UDP-Glc (**C**), UDP-GlcA (**D**), UDP-GlcNAc (**E**), and UDP-Gal (**F**). Receptor, ligands, and key residues are shown in cartoon and stick representations. Hydrogen bonds and salt bridges are displayed as red dashed lines. See *Supplementary file 2* for pairwise interaction details. (**G–I**) Concentration-response curves of calcium mobilization for UDP-GlcA (**G**), UDP-GlcNAc (**H**), and UDP-Gal (**I**) in HEK293 expressing P2Y14 wild-type (WT) and mutants. Data are shown as mean ± SEM (n=3). See also *Figure 3—source data 2*.

The online version of this article includes the following source data and figure supplement(s) for figure 3:

**Source data 1.** Potency of uridine diphosphate-glucuronic acid (UDP-GlcA), UDP-*N*-acetylglucosamine (GlcNAc), and UDP-galactose (Gal) in HEK293 expressing P2Y purinoceptor 14 (P2Y14).

**Source data 2.** Potency of uridine diphosphate-glucuronic acid (UDP-GlcA), UDP-*N*-acetylglucosamine (GlcNAc), and UDP-galactose (Gal) in HEK293

*Figure 3 continued*

expressing P2Y purinoceptor 14 (P2Y14) wild-type (WT) and mutants.

**Figure supplement 1.** Root mean square fluctuation (RMSF) values of ligand massive atomic position in simulations.

**Figure supplement 2.** Root mean square deviations (RMSD) values of ligand massive atoms as a function of time in simulations.

**Figure supplement 3.** Key interactions between P2Y purinoceptor 14 (P2Y14) and a uridine diphosphate (UDP)-sugar in simulations.

**Figure supplement 4.** Interactions between a key residue of P2Y purinoceptor 14 (P2Y14) and a uridine diphosphate (UDP)-sugar in simulations.

*C*). We docked ADP-Glc and ADP-Man to the X-ray structure of P2Y12 (*Zhang et al., 2014a*). The docking scores are –9.4 kcal/mol for ADP-Glc, for –10.0 kcal/mol for ADP-GlcA, and –9.3 kcal/mol for ADP-Man (*Figure 4—figure supplement 1B*), suggesting they stably bound to P2Y12. Consistently, in calcium mobilization assays, ADP-Glc, ADP-GlcA, and ADP-Man activated P2Y12 with EC50 values of 3.4±0.4 µM, 1.3±0.1 µM, and 12.3±0.9 µM, respectively (*Figure 4C*). Single-point mutations of K80$^{2.60}$,

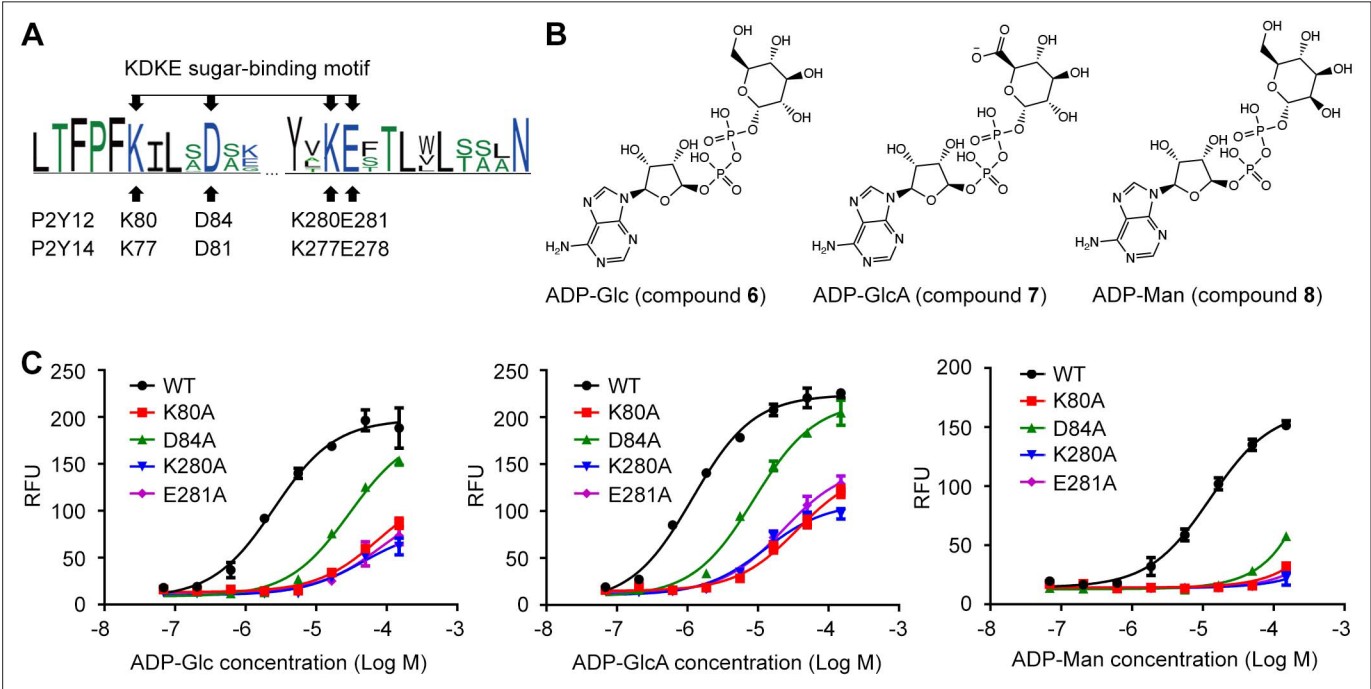

**Figure 4.** Adenosine diphosphate (ADP)-sugars binding to P2Y purinoceptor 12 (P2Y12). (**A**) Sequence log of the alignment between P2Y12 and P2Y14. Sequences of P2Y12 and P2Y14 involving 359 species were collected for making sequence alignments. See *Supplementary file 3* and *Figure 4—source data 1* for species repertoire information. The height of a letter is proportional to the relative frequency of that residue at a particular site. Four residues of K$^{2.60}$-D$^{2.64}$-K$^{7.35}$-E$^{7.36}$ (KDKE) sugar-binding motif are marked by arrows, with the corresponding residues in P2Y12 and P2Y14. (**B**) Chemical structure of ADP-glucose (Glc), ADP-glucuronic acid (GlcA), and ADP-mannose (Man). (**C**) Calcium mobilization concentration-response curves for ADP-Glc, ADP-GlcA, and ADP-Man in HEK293 expressing P2Y12 wild-type (WT) and mutants. Data are shown as mean ± SEM (n=3). See also *Figure 4—source data 1* and *Supplementary file 1*.

The online version of this article includes the following source data and figure supplement(s) for figure 4:

**Source data 1.** Potency of adenosine diphosphate-glucose (ADP-Glc), ADP-glucuronic acid (GlcA), and ADP-mannose (Man) in HEK293 expressing P2Y purinoceptor 12 (P2Y12) wild-type (WT) and mutants.

**Source data 2.** The neighbor joining tree of P2Y purinoceptor 12 (P2Y12) showing evolutionary range of species.

**Source data 3.** The neighbor joining tree of P2Y purinoceptor 13 (P2Y13) showing evolutionary range of species.

**Source data 4.** The neighbor joining tree of P2Y purinoceptor 14 (P2Y14) showing evolutionary range of species.

**Source data 5.** The neighbor joining tree of GPR87 showing evolutionary range of species.

**Figure supplement 1.** Conserved sugar-binding site on P2Y purinoceptor 12 (P2Y12).

**Figure supplement 2.** F277$^{7.32}$ as a key residue in sugar recognition of P2Y purinoceptor 12 (P2Y12).

**Figure supplement 3.** Conservation of each residue on P2Y purinoceptor 13 (P2Y13).

**Figure supplement 4.** Conservation of each residue on GPR87.

D84$^{2.64}$, K280$^{7.35}$, and E281$^{7.36}$ significantly diminished ADP-Glc-, ADP-GlcA-, and ADP-Man-induced responses, compared with WT P2Y12 (*Figure 4C*). These findings not only validate our assumption that P2Y12 can be activated by sugar nucleotides, but also indicate the conserved KDKE salt bridging chain as a functional motif for sugar binding. In addition, we found a less conserved phenylalanine residue F277$^{7.32}$, which is adjacent to the KDKE motif of P2Y12 (*Figure 4—figure supplement 2A*). The alanine substitution of F277$^{7.32}$ significantly reduced ADP-Glc-induced responses of P2Y12, compared with that of the WT group (*Figure 4—figure supplement 2B, C*). However, single-point mutations at the corresponding position of P2Y14 did not significantly affect the UDP-Glc-induced activation (*Figure 4—figure supplement 2C–F*). These findings indicate different nature of P2Y12 and P2Y14.

## Discussion

Mediated by UDP-sugars, P2Y14 plays an important role in immune responses and inflammation (*Arase et al., 2009*; *Barrett et al., 2013*; *Breton and Brown, 2018*; *Ferreira et al., 2017*; *Müller et al., 2005*; *Sesma et al., 2012*; *Sesma et al., 2016*), and possibly insulin resistance (*Wang et al., 2008*). Breton et al. found that kidney collecting duct intercalated cells present high levels of P2Y14, which is activated by UDP-Glc to promote neutrophil infiltration and renal inflammation (*Breton and Brown, 2018*). Exerting excessive P2Y14-mediated inflammatory reactions, high concentration of UDP-sugars was observed in extracellular tissue surrounding airway epithelial cells and lung secretions of cystic fibrosis patients (*Ferreira et al., 2017*; *Müller et al., 2005*; *Sesma et al., 2016*). UDP-Glc is also released from liver cells in obese states, possibly via hepatocellular apoptosis, leading to liver inflammation and insulin resistance (*Wang et al., 2008*). All these results indicate importance of UDP-sugar regulation of P2Y14 in pathological progresses.

In this work, we built the molecular model of UDP-Glc-bound P2Y14 to answer the long-standing question of sugar nucleotide regulation of the purinergic receptor. Binding to an extracellular pocket involving TMs 2 and 7 (*Figure 1*), the UDP-Glc might serve as intramolecular 'glue' attaching to TM6 and TM7 to activate P2Y14 (*Figure 2*). The agonist-induced remarkable conformational changes of TM6 and TM7 are also reported for P2Y12 (*Zhang et al., 2014a*). Compared with the AZD1283-bound (antagonist-bound) P2Y12 structure (*Zhang et al., 2014b*), the extracellular part of TM6 in the 2MeSADP-bound (agonist-bound) P2Y12 structure shifts over 10 Å and TM7 over 5 Å toward the center of TM helix bundle (*Zhang et al., 2014a*). The close parallel of P2Y12 and P2Y14 in the agonist-induced conformational changes indicates a common ligand-induced activation mechanism shared by purinergic receptors. In addition, in the studies involving the other UDP-sugars, we also found that the interactions between sugar-moieties of agonists with TM7 (E278$^{7.36}$) is determinant for UDP-sugars' potencies (*Figure 3*).

The carbohydrate-binding site has not been fully characterized for GPCRs. Except for P2Y14, it has not been reported that the other members of P2Y12-like subfamily can be directly activated by carbohydrates. Integrated computational modeling with mutagenesis study, we identified a conserved carbohydrate-binding motif (KDKE) for both P2Y14 and P2Y12 (*Figure 4*). The KDKE motif not only participates in receptor activation by bridging TM2 and TM7 (K77$^{2.60}$, D81$^{2.64}$, K277$^{7.35}$, and E278$^{7.36}$) (*Figure 2*), but also recognize different sugar moieties, including glucose, galactose, glucuronic acid, and *N*-acetylglucosamine groups (*Figure 3*, *Figure 4*). Remarkably, this KDKE motif can distinguish isomers as UDP-Glc and UDP-Gal. Our MD simulations showed that KDKE motif attracted the 6' hydroxyl group of glucose and interacted with the 4' hydroxyl group of galactose (*Figure 3C and F*). Consistent with our observations, a previous structure-activity relationship study revealed that selective mono-fluorination of the 6' hydroxyl group of the glucose moiety results to fourfold less potency on P2Y14 (*Ko et al., 2009*). As another member of P2Y12-like subfamily, P2Y13 also has the conserved KDKE site (*Figure 4—figure supplement 3*), suggesting it might be regulated by carbohydrates. GPR87 is a close homolog of P2Y14 with the sequence identity of 44.94%. GPR87 has K/R$^{2.60}$, D$^{2.64}$, K/E$^{7.35}$, and E$^{7.36}$ at the corresponding positions of the KDKE sugar-binding motif, indicating varied carbohydrate sensitivities of this receptor in different species (*Figure 4—figure supplement 4*). The species repertoires of these receptors consist of many amniotes but a few anamniotes (*Supplementary file 3a*). Distant relatives of the amniote receptors might show different sensitivities to carbohydrates. Similar to the KDKE motif of the receptors, the UDP-sugar-binding sites consisting of charged residues have been discovered for sugar transferases (*Gerlach et al., 2018*; *Hao et al., 2021*). In typical glycosylation transfers as TarP and SseK3, two aspartic acids and one positively

charged residue (arginine or lysine) participate in recognition of 3' or 4' hydroxyl groups of GlcNAc or GalNAc moiety (*Gerlach et al., 2018*; *Hao et al., 2021*). However, a salt bridging chain has not been observed in these sugar-binding sites. The different arrangements of UDP-sugar-binding sites between P2Y14 and these sugar transferases might be determinant for their sugar selectivity.

In conclusion, we revealed a conserved carbohydrate-binding motif in both P2Y12 and P2Y14, extending our understanding of how carbohydrates regulate GPCRs. Our molecular models of different sugar nucleotides provide great details for carbohydrate activation and recognition of these receptors, which would inspire further carbohydrate drug development for GPCRs. Whether the other carbohydrate-binding motifs exist in GPCRs is currently unknown. Further investigations focused on carbohydrate regulation of GPCRs will continue to add both new concepts and physiological understanding to the field.

# Materials and methods

**Key resources table**

| Reagent type (species) or resource | Designation | Source or reference | Identifiers | Additional information |
|---|---|---|---|---|
| Gene (*Homo sapiens*) | P2Y14 | GenBank | NM_001081455.2 | |
| Gene (*Homo sapiens*) | P2Y12 | GenBank | NM_022788.5 | Optimized |
| Strain, strain background (*Escherichia coli*) | *Trans*5α Chemically Competent Cell | TransGen Biotech | Cat.No: CD201-01 | |
| Cell line (*Homo sapiens*) | HEK293 | ATCC | CRL-1573 | |
| Antibody | Anti-HA primary antibody (Rabit monoclonal) | Cell Signaling Technology | Cat.No: 3724. | FCM (1:800) |
| Antibody | Goat anti-rabbit IgG(H+L) FITC conjugate secondary antibody (Goat monoclonal) | TransGen Biotech | Cat.No: HS111 | FCM (1:200) |
| Recombinant DNA reagent | pCDNA3-HA(plasmid) | This paper | | HA version of pCDNA3 |
| Sequence-based reagent | P2Y14-K77A-F | This paper | PCR primers | GACTTTTCCT TTCGCGATCC TTGGTGAC |
| Sequence-based reagent | P2Y14-K77A-R | This paper | PCR primers | GTCACCAAGG ATCGCGAAAG GAAAAGTC |
| Sequence-based reagent | P2Y14-D81A-F | This paper | PCR primers | CAAGATCCT TGGTGCCTC AGGCCTTGG |
| Sequence-based reagent | P2Y14-D81A-R | This paper | PCR primers | GACCAAGGCC TGAGGCACCAA GGATCTTG |
| Sequence-based reagent | P2Y14-N90A-F | This paper | PCR primers | GTCCCTGGCA GCTGGCCGTG TTTGTGTGCAG |
| Sequence-based reagent | P2Y14-N90A-R | This paper | PCR primers | CTGCACACAAA CACGGCCAGCT GCCAGGGAC |
| Sequence-based reagent | P2Y14-I170A-F | This paper | PCR primers | GAGGTTACACA AGCAAAATGTA TAGAACTG |
| Sequence-based reagent | P2Y14-I170A-R | This paper | PCR primers | GTTCTATACATT TTGCTTGTGTAA CCTC |

*Continued on next page*

*Continued*

| Reagent type (species) or resource | Designation | Source or reference | Identifiers | Additional information |
|---|---|---|---|---|
| Sequence-based reagent | P2Y14-R274A-F | This paper | PCR primers | CAAAAGAAA TCTTGGCGT ATATGAAAG AATTC |
| Sequence-based reagent | P2Y14-R274A-R | This paper | PCR primers | GAATTCTTTC ATATACGCCA AGATTTCTTTT G |
| Sequence-based reagent | P2Y14-K277A-F | This paper | PCR primers | CTTGCGGTAT ATGGCAGAAT TCACTCTG |
| Sequence-based reagent | P2Y14-K277A-R | This paper | PCR primers | CAGAGTGAAT TCTGCCATAT ACCGCAAG |
| Sequence-based reagent | P2Y14-E278A-F | This paper | PCR primers | GCGGTATAT GAAAGCATT CACTCTGCT AC |
| Sequence-based reagent | P2Y14-E278A-R | This paper | PCR primers | GTAGCAGAG TGAATGCTTT CATATACCG |
| Sequence-based reagent | P2Y12-K80A-F | This paper | PCR primers | CACATTCCC ATTCGCGAT CCTGTCAGA TG |
| Sequence-based reagent | P2Y12-K80A-R | This paper | PCR primers | CATCTGACAG GATCGCGAAT GGGAATGTG |
| Sequence-based reagent | P2Y12-D84A-F | This paper | PCR primers | CAAGATCCTGT CAGCTGCCAAG CTCGGTAC |
| Sequence-based reagent | P2Y12-D84A-R | This paper | PCR primers | GTACCGAGCTT GGCAGCTGACA GGATCTTG |
| Sequence-based reagent | P2Y12-F277A-F | This paper | PCR primers | GAGAACACTC TGGCCTACGT CAAGGAATC |
| Sequence-based reagent | P2Y12-F277A-R | This paper | PCR primers | GATTCCTTGAC GTAGGCCAGAG TGTTCTC |
| Sequence-based reagent | P2Y12-K280A-F | This paper | PCR primers | CTGTTCTACGT CGCGGAATCCA CATTG |
| Sequence-based reagent | P2Y12-K280A-R | This paper | PCR primers | CAATGTGGATT CCGCGACGTAG AACAG |
| Sequence-based reagent | P2Y12-E281A-F | This paper | PCR primers | GTTCTACGTCA AGGCATCCACA TTGTGGC |
| Sequence-based reagent | P2Y12-E281A-R | This paper | PCR primers | GCCACAATGTG GATGCCTTGAC GTAGAAC |
| Commercial assay or kit | KOD-plus-Ver.2 | TOYOBO | Cat.No: KOD-211 | |
| Commercial assay or kit | MycoBlue Mycoplasma Detector | Vazyme | Cat.No: D101-01 | |

| Reagent type (species) or resource | Designation | Source or reference | Identifiers | Additional information |
|---|---|---|---|---|
| Software, algorithm | GraphPad Prism 6 | GraphPad Prism 6 | | |

## Chemicals

UDP-GlcNAc was prepared from D-GlcNAc as reported previously (*Zheng et al., 2022*). UDP-Glc and UDP-GlcA were prepared from Sucrose (*Wang et al., 2022*). UDP-Gal was prepared from D-Gal (*Muthana et al., 2012*). ADP-Man was synthesized by a two-step strategy. In detail, Man-1-p was first synthesized from D-Man using NahK from *Bifidobacterium longum* (*Nishimoto and Kitaoka, 2007*) and ATP as phosphorylation donor. Man-1-p was purified from the reaction mixture by the silver nitrate precipitation method (*Wen et al., 2016*; *Wen et al., 2015*). Then, ADP-Man was synthesized from Man-1-p and ATP by a GDP-mannose pyrophosphorylase from *Pyrococcus furiosus*, which could take ATP as substrate.

## Cell lines

HEK293 were purchased from ATCC and the identity of the cell line was confirmed by carrying out fingerprinting (Shanghai Genening Biotechnologies Inc, Shanghai, China). In addition, HEK293 cells were tested using the MycoBlue Mycoplasma Detector Kit (Vazyme,China), indicating that the cells were not contaminated by *Mycoplasma.*

## Cell culture and transient transfections

HEK293 cells were cultured in Dulbecco's modified Eagle's medium with 10% fetal bovine serum. All cells were maintained at 37°C in humidified incubators with 5% $CO_2$ and 95% air. Human P2Y14 or P2Y12 receptors and G protein α-subunit ($G\alpha_{qi5}$) were transiently co-transfected with HEK293 cells using PolyJet In Vitro DNA Transfection Reagent (SignaGen) according to the manufacturer's instructions. Thus, a mixture of 1 μg of receptor DNA and 1 μg of $G\alpha_{qi5}$ DNA was used to transfect with the six-well plate cells at 90% confluency. HEK293 cells transiently expressing P2Y14 or P2Y12 receptor were subsequently used for the intracellular $Ca^{2+}$ assays after 48 hr post-transfection.

## Cell surface expression

Human P2Y14 or P2Y12 was cloned into a pcDNA3 vector with HA tag for expression in HEK293 cells. Mutants of P2Y14 or P2Y12 were constructed according to Fast Mutagenesis System (TransGen). Cell surface expression of P2Y14 or P2Y12 was analyzed by flow cytometry. HEK293 cells were transfected with pCDNA3-HA-P2Y14 or P2Y12 in six-well plate overnight. After having been incubated with rabbit anti-HA primary antibody (1:800, CST) for 1 hr at 4°C, the cells were incubated with goat anti-rabbit IgG(H+L) FITC conjugate secondary antibody (1:200, TransGen) for 50 min at 4°C. Data were collected with a flow cytometer (FACS Calibur, BD) and analyzed with FlowJo software.

## Intracellular $Ca^{2+}$ mobilization

Intracellular $Ca^{2+}$ assays were carried out as follows. HEK293 cells were seeded (80,000 cells/well) into Matrigel-coated 96-well plate 24 hr prior to assay. The cells were incubated with 2 μM Fluo-4 AM (Invitrogen) diluted in HBSS solution (0.4 g $L^{-1}$ KCl, 0.12 g $L^{-1}$ $Na_2HPO_4 \cdot 12H_2O$, 0.06 g $L^{-1}$ $KH_2PO_4$, 0.35 g $L^{-1}$ $NaHCO_3$, 0.14 g $L^{-1}$ $CaCl_2$, 0.10 g $L^{-1}$ $MgCl_2 \cdot 6H_2O$, 0.05 g $L^{-1}$ $MgSO_4$, and 8.0 g $L^{-1}$ NaCl) at 37°C for 50 min. After dye loading, the cells were treated with the compounds of interest. Then, calcium response (relative fluorescence unit) was measured using Flexstation 3 (Molecular Device) with fluorescence excitation made at 485 nm and emission at 525 nm.

## Molecular modeling, docking, and energy decomposition

Using the crystal structures of agonist-bound P2Y12 (PDB codes 4PXZ, 4PY0) (*Zhang et al., 2014a*) as templates, we employed Modeller (*Sali and Blundell, 1993*) to construct the human P2Y14 models. The human P2Y12 models are also built using these P2Y12 crystal structures (PDB codes 4PXZ, 4PY0) (*Zhang et al., 2014a*). The models with the lowest root mean square deviations from their template structures were selected for further analysis. A ligand was docked to the receptor using Schodinger Glide software in SP mode with default parameters (*Friesner et al., 2004*). A pocket binding to the

ligand with Glide G-scores below –6.5 kcal/mol was considered as a possible ligand-binding pocket. To involve receptor flexibility, we used RosettaLigand (*Davis and Baker, 2009*) to generate representative ligand-bound receptor models. After Rosetta-based docking, the top 1000 models with lowest binding energy score were selected. Then, they were further scored with the ligand-binding energy between ligand and receptor. The top 10 models with the lowest ligand-binding energy were selected for energy decomposition. The van der Waals energy of ligand binding was mapped to each receptor residue by residue_energy_breakdown utility (*Davis and Baker, 2009*). The model with the lowest ligand binding energy was used as the representative model.

## Modeling and simulations

To build a simulation system, we place the molecular model into a 1-palmitoyl-2-oleoyl-*sn*-glycero-3-phosphocholine lipid bilayer. The lipid-embedded complex model was solvated in periodic boundary condition box (80 Å × 80 Å × 120 Å) filled with TIP3P water molecules and 0.15 M KCl using CHARMM-GUI (*Wu et al., 2014*). Each system was replicated to perform three independent simulations. On the basis of the CHARMM36m all-atom force field (*Guvench et al., 2011*; *Huang et al., 2017*; *MacKerell et al., 1998*), MD simulations were conducted using GROMAS 5.1.4 (*Hess et al., 2008*; *Van Der Spoel et al., 2005*). After 100 ns equilibration, a 500 ns production run was carried out for each simulation. All productions were carried out in the NPT ensemble at temperature of 303.15 K and a pressure of 1 atm. Temperature and pressure were controlled using the velocity-rescale thermostat (*Bussi et al., 2007*) and the Parrinello-Rahman barostat with isotropic coupling (*Aoki and Yonezawa, 1992*), respectively. Equations of motion were integrated with a 2 fs time step; the LINCS algorithm was used to constrain bond length (*Hess, 2008*). Non-bonded pair lists were generated every 10 steps using distance cutoff of 1.4 nm. A cutoff of 1.2 nm was used for Lennard-Jones (excluding scales 1–4) interactions, which were smoothly switched off between 1 nm and 1.2 nm. Electrostatic interactions were computed using particle-mesh-Ewald algorithm with a real-space cutoff of 1.2 nm. The last 200 ns trajectory of each simulation was used to calculate average values.

## Sequence analysis

To analyze the conservation of residual sites, we collected sequences of receptors from UniProt database involving 359 species. Among these, we only found five anamniotes, including *Astyanax mexicanus*, *Xenopus tropicalis*, *Microcaecilia unicolor*, *Geotrypetes seraphini*, and *Xenopus laevis*. See *Supplementary file 3* for species repertoire information. The multiple sequence alignments were performed using Clustal Omega. Logplots generated for these alignments by WebLog. In each logplot, the height of a letter is proportional to the information content of an amino acid in bits, which was calculated by *Equation 1* as follows:

$$I = \log_2 N - \sum pi \log_2 pi \tag{1}$$

where $N$ is the number of all sequences and $pi$ is the probability of the amino acid in all sequences. A large value of the unit bits indicates a high conservation of a particular site.

## Statistics

Statistical analyses were performed using GraphPad Prism 6 (GraphPad Software). EC50 values for compounds were obtained from concentration-response curves by nonlinear regression analysis. Comparison of two compounds or two constructs was analyzed by unpaired t test to determine statistical difference. All statistical data are given as mean ± SEM of at least three independent experiments performed in duplicate or triplicate.

## Acknowledgements

This work was partially supported by Shanghai Municipal Science and Technology Major Project (to XC and LW), Lingang Laboratory grant (LG202102-01-01 to XC, LG-QS-202206-08 to LW), Fund of Youth Innovation Promotion Association (2022077 to XC), National Key Research and Development Program of China (2021YFA1301900 to XC), and National Natural Science Foundation of China (22007092 to LW).

## Additional information

### Funding

| Funder | Grant reference number | Author |
|---|---|---|
| Shanghai Municipal Science and Technology Major Project | | Xi Cheng |
| Lingang Laboratory grant | LG202102-01-01 | Xi Cheng |
| Lingang Laboratory grant | LG-QS-202206-08 | Liuqing Wen |
| Fund of Youth Innovation Promotion Association | 2022077 | Xi Cheng |
| National Key Research and Development Program of China | 2021YFA1301900 | Xi Cheng |
| National Natural Science Foundation of China | 22007092 | Liuqing Wen |

The funders had no role in study design, data collection and interpretation, or the decision to submit the work for publication.

### Author contributions

Lifen Zhao, Validation, Investigation, Methodology, Writing - original draft; Fangyu Wei, Methodology; Xinheng He, Visualization, Methodology; Antao Dai, Investigation, Methodology; Dehua Yang, Supervision, Investigation, Methodology; Hualiang Jiang, Supervision; Liuqing Wen, Supervision, Funding acquisition, Methodology, Writing - review and editing; Xi Cheng, Conceptualization, Supervision, Funding acquisition, Investigation, Methodology, Project administration, Writing - review and editing

### Author ORCIDs

Lifen Zhao ⬡ http://orcid.org/0009-0000-4158-790X
Dehua Yang ⬡ http://orcid.org/0000-0003-3028-3243
Xi Cheng ⬡ http://orcid.org/0000-0003-3735-645X

### Decision letter and Author response

Decision letter https://doi.org/10.7554/eLife.85449.sa1
Author response https://doi.org/10.7554/eLife.85449.sa2

## Additional files

### Supplementary files

• Supplementary file 1. Expression of mutants in HEK293.

• Supplementary file 2. Computational characterization of conformational changes and pairwise interactions of simulation models.

• Supplementary file 3. Species repertoire information for receptors.

• MDAR checklist

### Data availability

All data generated or analyzed during this study are included in the manuscript and supplement files.

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
