## [Editor Report]

This study describes a valuable model for the interaction of nucleotides, an important group of signalling molecules in health and disease, with their receptors. By combining experimental and theoretical methods, the authors provide compelling evidence for the mechanism of receptor activation. The results will be useful for drug design to target these receptors.

---

## [Decision Letter]

**Decision letter after peer review:**

Thank you for submitting your article "Identification of a carbohydrate-recognition motif of purinergic receptors" for consideration by *eLife*. Your article has been reviewed by 3 peer reviewers, one of whom is a member of our Board of Reviewing Editors, and the evaluation has been overseen by Kenton Swartz as the Senior Editor.

I apologize for the delay in handling your manuscript. It took some time to find the second reviewer and when reviewers 1 and 2 had submitted their reports, I was deeply involved in hosting a conference and other time-demanding matters at the end of the academic semester.

Essential Revisions (for the authors):

*Reviewing Editor (Recommendations for the authors):*

In addition to the comments provided by the other two reviewers, I wish to bring the authors' attention to the following matters:

1) The authors mention the sequence alignment repertoire of species on lines 360, 642-644, and in the legend to Figure 4A, but refer the reader to Supplementary file 3 for specific information apart from the fact that 379 species were included. It is far more important to describe the evolutionary range of species than the number of species, so I urge the authors to provide this information in the text. As far as I can see, the list in the supplementary file includes large numbers of mammals and birds and some turtles, lizards, and a snake, thus a panel consisting of amniotes. A quick look identified no amphibians nor any more distant relatives of the amniotes such as rayfinned fishes or cartilaginous fishes, although my tentative exploration of the members of this receptor family indicates that they existed before the origin of vertebrates. If the species repertoire is restricted to amniotes, this must be clearly stated in the text in the places discussing alignment and sequence conservation.

2) Another aspect of the alignment displayed in Figure 4 is that some positions are extremely difficult to read, for instance, positions 1, 2, 3, 4, and 266. The less frequent residues at many positions are totally impossible to see. But it may suffice to highlight the predominant residue at each position in this way if the authors do not discuss any of the secondary or tertiary amino acids that occur at a certain position.

3) Why does the alignment in Figure 4 use both capital F and small f? All other amino acid abbreviations use capital letters as far as I can see. Position 34 has both F and f – what is the difference? This may be explained somewhere in the text, but if so I must have missed it.

*Reviewer #1 (Recommendations for the authors):*

I will proceed with a detailed list of suggestions and criticisms following the article reading, irrespective of minor/major suggestions, the last of which will have an easy match for the authors with the weaknesses identified in the public review:

1) Line 59 – typo in the full name of 2MeSADP. Line 61 – please mention EL2 as part of the binding site for this ligand, it makes important interactions.

2) Lines 74-77 – Figure 1B and Figure 1H show a decrease in max EFFICACY for some ligands (UDP-Glc, 1H; K77A and D81A for the UDP-Glc mutations, panel of 1H). The way this is presented is confusing, I would classify UDP as a PARTIAL agonist as compared to UDP-Glc based on this graph. The implications of the effect of the mutations on UDP-Glc are however more delicate, it seems that these residues are involved in the activation of the receptor since the mutation leads to partial efficacy (reduction of the max efficacy) for this ligand. You must use the appropriate pharmacological parameters (Bmax) for presenting and discussing these results.

3) Lines 84-87 – docking scores are not sensible enough to prove this affirmation. The stability of binding has nothing to do with a score of docking, which can only be used to rank docking poses and, in the best cases, to distinguish good and bad binders (according to this data, both UDP and UDP-Glc bind equally well, which correlates with the experimental data and that is as far as you can go).

4) Lines 105 – The effect of a mutation like E278A on the potency (as measured in functional assay) of UDP-Glc is as low as 1.5 fold, which is far from being significant, despite the P value to compare with UDP, and in this sense, I don't see a proof for the specific role of this residue in the sugar binding. K77 and D81 definitely have a role which, despite EC50, modifies Bmax (see my comment below, you must discuss the last value)

5) The MD trajectories are minimally analyzed. The salt bridge analysis and inter-helical distances (Supplementary file 2-appendix table 1) are fine. But for the proposed H-bonds, only the distance between each of the four residues here postulated as forming the carbohydrate-binding site (K277, E278, R253, K77) are monitored with respect to the minimum distance to a sugar atom, (Figure 3 Figure supplement 2). It is well known that the occurrence of a H Bond depends not only on the distance but also on the geometry, so the percentage of occurrence of a H Bond should be reported. In addition, RMSF of the ligand would be informative as to the stability of the binding pose (even if it is an evolved binding pose from the initial docking, we need to verify that this is stable).

6) In this sense, it is stated (l 168 – 170) that E278 causes an electrostatic repulsion only on the UDP-GlcA derivative and this explains the increase in efficacy of this molecule upon E278A mutation. I don't see such a selective repulsion as all molecules except UDP-Glc show the same increased distance to this residue, and as happened before, the experimental data is again overinterpreted since E278A (Figure 3G, compare purple vs black curves) does not enhance the activity of UDP-GlcA in a meaningful way (60 vs 38 nM EC50, p<0.01).

7) The same applies to the discussion of the binding mode of UDP-GlcNAc (l177, Figure 3H). A selective interaction is postulated with D81 based on MD going away from the initial interaction with E278A; but indeed alanine mutation of both residues affects equally. Please modify your interpretation of the data accordingly.

8) Generally speaking, I am convinced that the carbohydrates binding site identified is consistent with the mutagenesis data, but the discussion of specific interactions for each molecule goes beyond what can be demonstrated by the data, and one should draw general lines when not certain of specific interactions, or clearly state these are hypothesis, not all of them backed up by the experimental data.

9) P2Y12 data (Figure 4) – It seems clear that the conserved positions for the four amino acids discussed affect the ADP-sugar activation on P2Y12, but at any point, the authors comment the much lower potencies of the WT for the sugar-derivatives of ADP on this receptor, as compared to the UDP-sugars on P2Y12. The nature of the binding site of the two receptors can probably explain why this difference in potency.

l250 – incomplete sentence, modify accordingly.

*Reviewer #2 (Recommendations for the authors):*

1. As the author noted in line 117, the molecular docking process only considers the rigidity of the receptor, thereby excluding the receptor's flexibility. To provide a more comprehensive understanding of the system's dynamic properties and validate the docking results, the use of the representative binding pose from molecular dynamic simulations is commendable. Furthermore, it is noteworthy that K77 is relatively far away from the UDP in the present predicted binding pose, despite functional data indicating its importance. MD analysis may offer some insights into this discrepancy.

2. In Figure G, the residue contribution is solely based on the docking results. However, using MM/PB(GB)SA for a similar decomposition analysis can yield more precise outcomes. As such, it would be worthwhile to perform this analysis using the MD trajectory.

3. Having only a single replica with 500 ns of simulation may not provide enough sampling to achieve meaningful results. It is recommended that the author provides the agonist RMSD vs time to assess the convergence of the simulations. Furthermore, performing at least three replicas can increase the statistical significance of the findings and further validate the results.

---

## [Author Response]

Essential Revisions (for the authors):Reviewing Editor (Recommendations for the authors):In addition to the comments provided by the other two reviewers, I wish to bring the authors' attention to the following matters:1) The authors mention the sequence alignment repertoire of species on lines 360, 642-644, and in the legend to Figure 4A, but refer the reader to Supplementary file 3 for specific information apart from the fact that 379 species were included. It is far more important to describe the evolutionary range of species than the number of species, so I urge the authors to provide this information in the text. As far as I can see, the list in the supplementary file includes large numbers of mammals and birds and some turtles, lizards, and a snake, thus a panel consisting of amniotes. A quick look identified no amphibians nor any more distant relatives of the amniotes such as rayfinned fishes or cartilaginous fishes, although my tentative exploration of the members of this receptor family indicates that they existed before the origin of vertebrates. If the species repertoire is restricted to amniotes, this must be clearly stated in the text in the places discussing alignment and sequence conservation.

We appreciate your comments and have updated the sequence alignment repertoire of species for P2Y12, P2Y13, P2Y14 and GPR87 in Supplementary file 3. The gene names of these receptors (P2YR12, P2YR13, P2YR14 and GPR87) were used to collect protein sequences in the UniProt database. If a species had multiple entries for the same protein, the entry with the longest sequence and the evidence for protein existence was selected. The updated Supplementary file 3a includes 359 species, which are common in P2Y12, P2Y13, P2Y14 and GPR87. To describe the evolutionary range of species, we generated neighbor joining trees for these 359 species of each receptor (source data for Figure 4) in the revised manuscript.

It is true that large numbers of amniotes are included in the species repertoires. In fact, among the 359 common species (Supplementary file 3a), we only found five anamniotes, including *Astyanax mexicanus*, *Xenopus tropicalis, Microcaecilia unicolor, Geotrypetes seraphini,* and *Xenopus laevis.* We analyzed species for each receptor. P2Y12, P2Y13, P2Y14 and GPR87 involve 420, 419, 460 and 409 species, respectively. For anamniotes, P2Y12, P2Y13, P2Y14 and GPR87 only have 14, 20, 56 and 14 protein sequences, respectively. We further analyzed the sequence alignments for anamniote receptors. As shown in Author response images 1-4, Figure 4—figure supplement 1, Figure 4—figure supplement 3 and Figure 4—figure supplement 4, the KDKE motif are conserved in the 359 common species, but less conserved in the anamniotes. Therefore, distant relatives of the amniote receptors might show different sensitivities to carbohydrates. We have included this information in the text in the place discussing alignment and sequence conservation in the revised manuscript.

**Author response image 1. sa2fig1:** Conservation of each residue on P2Y12 in 14 anamniotes. The height of a letter is proportional to the relative frequency of that residue at a particular site. Residues of KDKE sugar-binding motif are labeled with red arrows. See Supplementary file 3 for species repertoire information.

**Author response image 2. sa2fig2:** Conservation of each residue on P2Y13 in 20 anamniotes. The height of a letter is proportional to the relative frequency of that residue at a particular site. Residues of KDKE sugar-binding motif are labeled with red arrows. See Supplementary file 3 for species repertoire information.

**Author response image 3. sa2fig3:** Conservation of each residue on P2Y14 in 359 common species (*left*) and 56 anamniotes (*right*). The height of a letter is proportional to the relative frequency of that residue at a particular site. Residues of KDKE sugar-binding motif are labeled with red arrows. See Supplementary file 3 for species repertoire information.

**Author response image 4. sa2fig4:** Conservation of each residue on GPR87 in 14 anamniotes. The height of a letter is proportional to the relative frequency of that residue at a particular site. Residues of KDKE sugar-binding motif are labeled with red arrows. See Supplementary file 3 for species repertoire information.

2) Another aspect of the alignment displayed in Figure 4 is that some positions are extremely difficult to read, for instance, positions 1, 2, 3, 4, and 266. The less frequent residues at many positions are totally impossible to see. But it may suffice to highlight the predominant residue at each position in this way if the authors do not discuss any of the secondary or tertiary amino acids that occur at a certain position.

Thanks for your comments. We have remade all alignment plots by highlighting the predominant residue at each position in the revised manuscript (Figure 4—figure supplements 1, 3, 4).

3) Why does the alignment in Figure 4 use both capital F and small f? All other amino acid abbreviations use capital letters as far as I can see. Position 34 has both F and f – what is the difference? This may be explained somewhere in the text, but if so I must have missed it.

Thanks for your comments. We have remade all alignment plots by employing consistent representations for residues in the revised manuscript (Figure 4—figure supplements 1, 3, 4).

Reviewer #1 (Recommendations for the authors):I will proceed with a detailed list of suggestions and criticisms following the article reading, irrespective of minor/major suggestions, the last of which will have an easy match for the authors with the weaknesses identified in the public review:1) Line 59 – typo in the full name of 2MeSADP. Line 61 – please mention EL2 as part of the binding site for this ligand, it makes important interactions.

The full name of 2MeSADP is 2-methylthio-adenosine-5'-diphosphate. And we agree that the extracellular loops including ECL2 are part of the binding site for this ligand. We have revised the manuscript to correct the typo and include this information.

2) Lines 74-77 – Figure 1B and Figure 1H show a decrease in max EFFICACY for some ligands (UDP-Glc, 1H; K77A and D81A for the UDP-Glc mutations, panel of 1H). The way this is presented is confusing, I would classify UDP as a PARTIAL agonist as compared to UDP-Glc based on this graph. The implications of the effect of the mutations on UDP-Glc are however more delicate, it seems that these residues are involved in the activation of the receptor since the mutation leads to partial efficacy (reduction of the max efficacy) for this ligand. You must use the appropriate pharmacological parameters (Bmax) for presenting and discussing these results.

We appreciate the above comments. As the reviewer suggested, we performed additional radiolabeled ligand binding assay, try to measure the binding parameters of P2Y14 with UDP or UDP-Glc. Unfortunately, we could not obtain a specific binding signal using the ^3^H-labeled uridine diphosphate D-glucose (^3^H-UDP-Glc, PerkinElmer, #NET1163250UC). The reported literature also showed no reasonable specific binding window when using ^3^H-UDP-Glc as radioligand *(J Biomol Screen 2011, 16(9):1098*). Based on the current results of intracellular ca^2+^ mobilization, we agree with the reviewer’s comment that UDP is a partial agonist as compared to UDP-Glc which maybe attribute to the extra binding sub-pocket2 by UDP-Glc and require further validation.

3) Lines 84-87 – docking scores are not sensible enough to prove this affirmation. The stability of binding has nothing to do with a score of docking, which can only be used to rank docking poses and, in the best cases, to distinguish good and bad binders (according to this data, both UDP and UDP-Glc bind equally well, which correlates with the experimental data and that is as far as you can go).

We agree with your comments. The docking score of UDP-Glc is -9.3 kcal/mol and that of UDP is -8.5 kcal/mol, indicating that both UDP and UDP-Glc bind to P2Y14. Accordingly, we have modified the statement in the revised manuscript.

4) Lines 105 – The effect of a mutation like E278A on the potency (as measured in functional assay) of UDP-Glc is as low as 1.5 fold, which is far from being significant, despite the P value to compare with UDP, and in this sense, I don't see a proof for the specific role of this residue in the sugar binding. K77 and D81 definitely have a role which, despite EC50, modifies Bmax (see my comment below, you must discuss the last value)

We agree with your comments and removed the statement that the effect of a mutation (like E278A) on the potency of UDP-Glc is significant in the revised manuscript.

5) The MD trajectories are minimally analyzed. The salt bridge analysis and inter-helical distances (Supplementary file 2-appendix table 1) are fine. But for the proposed H-bonds, only the distance between each of the four residues here postulated as forming the carbohydrate-binding site (K277, E278, R253, K77) are monitored with respect to the minimum distance to a sugar atom, (Figure 3 Figure supplement 2). It is well known that the occurrence of a H Bond depends not only on the distance but also on the geometry, so the percentage of occurrence of a H Bond should be reported. In addition, RMSF of the ligand would be informative as to the stability of the binding pose (even if it is an evolved binding pose from the initial docking, we need to verify that this is stable).

Thanks for the valuable comments. We calculated the percentage of occurrence of hydrogen binding between the ligand and the carbohydrate-binding site (K277, E278, R253 and K77) (Supplementary file 2c) for the proposed hydrogen bonds. We also calculated the ligand RMSF to show the stability of the ligand-binding pose in MD simulations (Figure 3—figure supplement 1). These results have been added in the revised manuscript.

6) In this sense, it is stated (l 168 – 170) that E278 causes an electrostatic repulsion only on the UDP-GlcA derivative and this explains the increase in efficacy of this molecule upon E278A mutation. I don't see such a selective repulsion as all molecules except UDP-Glc show the same increased distance to this residue, and as happened before, the experimental data is again overinterpreted since E278A (Figure 3G, compare purple vs black curves) does not enhance the activity of UDP-GlcA in a meaningful way (60 vs 38 nM EC50, p<0.01).

We agree with your comments. For UDP-Glc, both K277^7.35^ and E278^7.36^ could form hydrogen bonds with the glucose 6’ hydroxyl group to keep TM7 close to the receptor core in MD simulations (Figure 3C, Figure 3—Figure supplement 4A). However, in UDP-GlcA-bound P2Y14 simulations, at the corresponding position, the 5’ carboxyl group of the sugar moiety failed to form hydrogen bond with the negative charged E278^7.36^ (Figure 3D, Figure 3—Figure supplement 4B). Compared with that of WT group (EC50 = 59.9 ± 4.8 nM), the single-point mutation of E278A did not decrease the UDP-GlcA-induced calcium mobilization (EC50 of 38.2 ± 2.2 nM) (Figure 3G), supporting the proposed sugar-binding model (Figure 3D). Accordingly, we have modified the statement in the revised manuscript.

7) The same applies to the discussion of the binding mode of UDP-GlcNAc (l177, Figure 3H). A selective interaction is postulated with D81 based on MD going away from the initial interaction with E278A; but indeed alanine mutation of both residues affects equally. Please modify your interpretation of the data accordingly.

We agree with your comments. Single-point alanine mutation of both D81 and E278 significantly reduce the UDP-GlcNAc-induced receptor activities (Figure 3H). Accordingly, we have modified the statement in the revised manuscript.

8) Generally speaking, I am convinced that the carbohydrates binding site identified is consistent with the mutagenesis data, but the discussion of specific interactions for each molecule goes beyond what can be demonstrated by the data, and one should draw general lines when not certain of specific interactions, or clearly state these are hypothesis, not all of them backed up by the experimental data.

Thanks for your comments. In the revised manuscript, we have removed or modified the statements that are not fully backed up by the experimental data.

9) P2Y12 data (Figure 4) – It seems clear that the conserved positions for the four amino acids discussed affect the ADP-sugar activation on P2Y12, but at any point, the authors comment the much lower potencies of the WT for the sugar-derivatives of ADP on this receptor, as compared to the UDP-sugars on P2Y12. The nature of the binding site of the two receptors can probably explain why this difference in potency.

Discovery of highly potent P2Y12 agonists requires screening of a large number of compounds. In this work, we only designed, synthesized and tested three ADP-sugars (ADP-Glc, ADP-GlcA and ADP-mannose) to validate our assumption that sugar nucleotides can activate human P2Y12. It is possible there are the other ADP-sugars, which are highly potent P2Y12 agonists. It is technically challenging to synthesize ADP-sugars. Currently, we can only obtain ADP-Glc, ADP-GlcA and ADP-Man. Once the other ADP-sugars are available for us, we will test them and try to discover highly potent agonists in the future work.

To investigate the different nature of the binding site of P2Y12 and P2Y14, we identified 22 sugar nucleotide-binding positions in two receptors, including 16 highly conserved positions and six less conserved positions (Author response image 5). Among these less conserved ones, the F^7.32^ (F277 at 7.32 position) of human P2Y12 is adjacent to the KDKE motif (Figure 4—figure supplement 2A). The alanine substitution of F^7.32^ significantly reduced ADP-Glc-induced calcium mobilizations of human P2Y12, compared with that of the WT group (Figure 4—figure supplement 2B, C). But the mutations of the corresponding residue of human P2Y14 (R274A or R274F) did not significantly affect the UDP-Glc-induced responses (Figure 4—figure supplement 2D-F). These results suggest the phenylalanine at 7.32 position of human P2Y12 might contribute to the interaction with ADP-Glc, while the residue at the corresponding position of human P2Y14 might not participate in binding of UDP-Glc. This is consistent with the assumption that the nature of the binding site of the two receptors is different. We have included this information in the revised manuscript.

**Author response image 5. sa2fig5:** Sequence log of the alignment between P2Y14 and P2Y12. Sequences of P2Y14 and P2Y12 involving 359 species were collected for making sequence alignments. See Supplementary file 3 for species repertoire information. The height of a letter is proportional to the relative frequency of that residue at a particular site. Sugar nucleotide-binding residues are highlighted in yellow. Four residues of KDKE sugar-binding motif are marked by arrows, with the corresponding residues in P2Y14 and P2Y12.

l250 – incomplete sentence, modify accordingly.

Thank you for pointing this out. We have modified the sentence in the revised manuscript accordingly.

Reviewer #2 (Recommendations for the authors):1. As the author noted in line 117, the molecular docking process only considers the rigidity of the receptor, thereby excluding the receptor's flexibility. To provide a more comprehensive understanding of the system's dynamic properties and validate the docking results, the use of the representative binding pose from molecular dynamic simulations is commendable. Furthermore, it is noteworthy that K77 is relatively far away from the UDP in the present predicted binding pose, despite functional data indicating its importance. MD analysis may offer some insights into this discrepancy.

We performed three independent MD simulations for the UDP-bound P2Y14 system. The system setup is same as the UDP-Glc-bound P2Y14 simulation system. The average minimal distance between the UDP and K77 was 5.9 ± 0.1 Å (Author response image 6), consisting with the docking model. This result suggests that K77 does not stably bind to UDP to affect the receptor activation.

**Author response image 6. sa2fig6:** Binding of UDP to P2Y14 in MD simulations. (A) A representative simulation model showing UDP in the sub-pocket of P2Y14. Key residues are highlighted in yellow. Two sub-pockets for ligand binding are marked with circles. (B) Massive atom distance between UDP and K77 in three independent simulations of UDP-bound P2Y14. Three replicated simulations were indicated by different colors.

2. In Figure G, the residue contribution is solely based on the docking results. However, using MM/PB(GB)SA for a similar decomposition analysis can yield more precise outcomes. As such, it would be worthwhile to perform this analysis using the MD trajectory.

We performed MMPBSA calculation using the MD trajectory of UDP-Glc-bound P2Y14 with implicit membrane. The calculation followed the Amber 20 MMPBSA.py protocol, and as in our manuscript, we used the last 200 ns of the trajectories. The ligand-binding energy was decomposed for each receptor residue (Author response image 7). The total contribution of two negatively charged residues D81 and E278 are more than 25 kcal/mol, suggesting they made negative contributions to UDP-Glc binding. This is inconsistent with our experimental observations that D81 and E278 are key residues for UDP-Glc-induced activation of P2Y14 (Figure 1H). It's important to note that MMPBSA uses an implicit solvent model and thus may not effectively capture solvent-induced interactions between sugar-moiety and charged residues. This discrepancy might explain the deviations between the calculation outcomes and experimental results.

To further characterize the ligand-binding pose, we calculated the percentage of occurrence of hydrogen binding between the ligand and the carbohydrate-binding site (K277, E278, R253 and K77) for the proposed H-bonds (Supplementary file 2c). We also calculated the ligand RMSF to show the stability of the ligand-binding pose in MD simulations (Figure 3—figure supplement 1). These results have been added in the revised manuscript.

**Author response image 7. sa2fig7:** Decomposition of UDP-Glc-binding contribution for each receptor residue using MMPBSA.

3. Having only a single replica with 500 ns of simulation may not provide enough sampling to achieve meaningful results. It is recommended that the author provides the agonist RMSD vs time to assess the convergence of the simulations. Furthermore, performing at least three replicas can increase the statistical significance of the findings and further validate the results.

We performed three replicas for each simulation system in our work as mentioned in the method section. To clarify this, we added simulation information in the main text in the revised manuscript. We calculated the agonist RMSDs vs time to show the convergence of the simulations (Figure 3—figure supplement 2). This information is also included in the revised manuscript.